# Novel Antireflection Coatings Obtained by Low-Temperature Annealing in the Presence of Tetrabutylammonium Bromide and Gold Nanoparticles

**DOI:** 10.3390/ma15217658

**Published:** 2022-10-31

**Authors:** Alena A. Lokteva, Anastasiia A. Kotelnikova, Roman S. Kovylin, Alexey N. Konev, Alexandr V. Piskunov

**Affiliations:** G.A. Razuvaev Institute of Organometallic Chemistry, Russian Academy of Sciences, 603950 Nizhny Novgorod, Russia

**Keywords:** sol compositions, antireflection coatings, tetrabutylammonium bromide, gold nanoparticles, optical properties

## Abstract

In this work, nanoporous antireflective coatings on silicate glass were obtained from silicon dioxide sol compositions by the sol-gel method in the presence of quaternary ammonium salt (tetrabutylammonium bromide) at different annealing temperatures (200–250 °C). Varying the salt concentration from 3 to 5 wt.%, we achieved the transmittance of the coatings of about 97% at 250 °C in comparison with 91% for clean glass in the wavelength range from 400 to 1100 nm. The addition of gold nanoparticles to the composition containing 5 wt.% tetrabutylammonium bromide allowed us to decrease the annealing temperature to 200 °C, preserving the transmittance at the level of 96.5%. For this case, the optimal concentration of gold nanoparticles is determined (2.6 × 10^−9^ mol/mL). According to the SEM analysis, the obtained antireflective coatings contain pores with a minimum area size up to 4 nm^2^.

## 1. Introduction

Recently, much attention has been paid to the development of antireflective, self-cleaning, antifreeze and other types of coatings on silicate glass. Glass with antireflective coatings are in high demand for solar panels, LED modules, greenhouses and other applications [1,2,3,4,5,6,7,8].

In general, the reflection of light occurs at the interface between two phases: glass and air, which differ in the refractive index. Supposing that the refractive index of optical glass is 1.5 and the air is 1.0, according to the interference principle and Fresnel formula, the theoretical refractive index of the single-layer antireflection coating should be in the range from 1.20 to 1.25 [9]. To ensure a high transmittance of light, low-refractive nanoporous single-layer films are applied to the glass. For example, the literature discusses optical single-layer quarter-wave antireflection coatings based on nanoporous silicon dioxide obtained from sol compositions of a special chemical content, and also considers the technological parameters of the coating process and the temperature regime of gel annealing [10,11,12,13,14,15,16,17,18].

The sol-gel method is the easiest and most promising way to make antireflection coatings because of the possibility to adapt the microstructure of the applied film. Porosity reduces the refractive index and maximizes the coated glass transmittance. For the matrix refractive index n ~ 1.46 (in the case of SiO_2_), the porosity of 0.45–0.50 ensures n ~ 1.23 of a single-layer anti-reflective coating [19]. For this purpose, various pore-generating agents, such as surfactants, are used [20,21]. Determining the optimum gel temperature and annealing time is one of the challenges in achieving antireflective parameters on silicate glass. As a rule, coated glasses with a good antireflection effect (~95–99%) are obtained at annealing temperatures of T = 400–500 °C, which is an energy-consuming and economically unprofitable process, especially on an industry. Therefore, the problem of obtaining transparent coatings on silicate glass from silicon dioxide at a curing temperature of 200–250 °C is an urgent one.

Previously, the preparation of films based on mesoporous silicon dioxide and cationic surfactant cetyltrimethylammonium bromide (CTAB) was reported [22,23]. We prepared such films to make anti-reflective coatings on silicate glasses and investigated the effect of the CTAB concentration in SiO_2_ solution on the transmittance of the coated glass in the wavelength range of 200–1100 nm. The ultimate transmittance of ~99% has been achieved for nanoporous antireflection coatings prepared at an annealing temperature of 500 °C and a surfactant concentration in the sol of 5.2 × 10^−2^ mol/L (1.9 wt.%). CTAB belongs to the quaternary ammonium salts and is a cationic surfactant. Due to its prolonged alkyl tail, CTAB forms various agglomerates (spherical and rod micelles, hexagonal, cubic, and lamellar liquid crystal structures) under increased concentrations in water. The formation of a thin film from a solution of silica with CTAB and subsequent heating to 500 °C (for thermal destruction of surfactants) generates a regular cubic mesophase [24].

In this work, we chose another quaternary ammonium salt, tetrabutylammonium bromide (TBAB) to study its effect on the pore structure of the antireflective film. Unlike CTAB, this salt is not able to form regular structures, since it is not a surfactant and has a shorter alkyl tail. On the other hand, the presence of TBAB changes the pore structure and can enhance the transmittance of the coating at lower annealing temperatures. It is known that the introduction of gold particles into an alcoholic medium provides the thermal destruction of organics [25,26,27,28,29,30,31,32]. So, we report the introduction of a new additive, a quaternary ammonium salt, into a sol-gel composition and the effect of its concentration on the film transparency. For the first time, we present the results of using gold nanoparticles as a catalyst for the decomposition of an organic additive in such systems.

## 2. Materials and Methods

### 2.1. Preparation of Silicon Dioxide Sol

Silica sol was prepared by hydrolysis of tetraethoxysilane (TEOS). The mixture of 44.7 mL TEOS (41.6 g), 5.4 mL distilled water and 9.0 mL 0.1 N hydrochloric acid solution was placed in a 100 mL glass flask and stirred on a magnetic stirrer for 0.5 h. Then, 40.8 mL of isopropyl alcohol was added. The concentration of silica sol obtained was 2 mol/L. Dilution with isopropyl alcohol gives a sol concentration of 0.3 mol/L. The average size of silicon dioxide nanoparticles is ~2 nm. Then, 3 or 5 wt.% TBAB was added under constant stirring with a magnetic stirrer. The resulting composition was applied to a glass substrate by dipping. The nanocomposite films were annealed at temperatures of 200 or 250 °C.

### 2.2. Method of Obtaining Colloidal Gold Nanoparticles

Colloidal gold nanoparticles were obtained by the method described in [33]. We placed 500 mL of HAuCl_4_·4H_2_O (solution) 1 mM in a 1 L round-bottom flask with a reflux condenser. It was brought to a boil with vigorous stirring, and then 50 mL of 38.8 mM sodium citrate was quickly added. The color of the solution quickly changed from yellow to dark red. The solution was refluxed for another 10 min. Then, the heating was removed, and the reaction mixture was stirred for another 15 min. The solution was cooled and filtered. The average size of the gold nanoparticles is 1–2 nm.

### 2.3. The Method of Obtaining the Sol-Composition with the Addition of Colloidal Gold Nanoparticles

Sol compositions were prepared in the presence of TBAB (3 and 5 wt.%) according to the procedure described above. Then, the solution of gold nanoparticles was added to the sol-composition (the concentration was varied from 0.9 × 10^−9^ to 4.4 × 10^−9^ mol/mL).

### 2.4. Preparation of a SiO_2_ Xerogel Film on Silicate Glass

Optical microscopy silicate glasses 25 mm × 75 mm × 1 mm were used as a substrate. The refractive index of the glass is 1.506. The glass surface was cleaned of contaminants by immersion in a hydrogen peroxide alkaline solution, and then the glass was washed with distilled water and dried in the thermostat at 150 °C for 6–8 h. The coatings were applied to the glass by dipping (at a temperature of 20–25 °C and a humidity of 40–60%). This was done with a custom-designed laboratory coating device. The device consists of a high-precision speed reducer mechanism supported with a computer-controlled stepping motor. The glass was immersed in a bath containing the sol composition, and after that the glass was lifted from the bath at a constant speed. The coated glass was kept at room temperature for 1 h to remove most of the volatile compounds from the coatings. Then, the glass was heated to 200 or 250 °C and held at this temperature for 60 min. The transmittance of the applied film was measured in the wavelength range from 200 to 1100 nm on the Perkin-Elmer Lambda 25 spectrometer. The thickness and refractive index of the film coatings were determined using the LEF-3M1 ellipsometer.

The size of silica nanoparticles in solution and the size of gold nanoparticles were analyzed by dynamic light scattering (DLS) on the NanoBrook Omni instrument (Brookhaven Instruments, Holtsville, NY, USA) using a solid-state laser (660 nm). DLS experiments were performed at an angle of 90°. Signal analyzer was used in multimodal mode. The accuracy of temperature control in the cell containing the nanoparticle suspension was ±0.1 °C.

Thermogravimetric analysis (TGA) of TBAB was performed on the Perkin Elmer Pyris 6 TGA analyzer in an air atmosphere (heating rate 5 deg min^−1^, flow rate 80 mL min^−1^).

The surface of antireflection coatings on silicate glass was investigated using the Regulus SU8100 scanning electron microscope (Hitachi, Tokyo, Japan) equipped with XFlash^®^ 6|60 EDS (Bruker, Bremen, Germany) energy dispersive microanalysis (EDS) system. A sample of 3 mm × 3 mm was without any conductive coating. The structure of the antireflection coatings was analyzed and EDS was performed at an accelerating voltage of 15 kV, which allowed a wide range of elements to be detected.

Histograms of pore area distribution for the studied samples were obtained by SEM image processing in the ImageJ program.

## 3. Results and Discussion

Figure 1 shows the transmittance curves for uncoated glass (curve 1), and for glass with double-sided single-layer silica coating obtained from silica solution with TBAB in concentrations of 3 and 5 wt.% at an annealing temperature of 200 °C (curves 2, 3, respectively) and 250 °C (curves 4, 5). The maximal transmittance of the initial silicate glass is ~91 ± 0.3% at 520 nm (curve 1). The coating enhances optical properties of the glass in the wavelength range from 400 to 1100 nm. However, this effect is insignificant at the 200 °C annealing temperature. Turning the TBAB concentration from 3 to 5 wt.% results in a slight increase in transmittance of the sample in the long wavelength spectrum. At this temperature, TBAB remains in the coating, which is confirmed by the thermogravimetric analysis (Figure 2). The temperature of 200 °C corresponds to the initial stage of TBAB decomposition. Additionally, at the annealing temperature of 250 °C, the additive is decomposed completely (Figure 2), which results in the transmittance improvement. The best effect is observed when 5 wt.% TBAB was added to the silica solution, and the maximum transmittance of the coated glass increased to ~97.5 ± 0.3% at the wavelength of ~550 nm (Figure 1, curve 5). In the mixture, the optimal content of the additive is determined by the ratio of silicon dioxide and additive concentrations. We assume that the greater additive content in the solution, the greater the size of nanopores and their total volume in the annealed coating. Accordingly, the refractive index becomes smaller and the optical transmittance of the coated glass increases.

Next, we studied the effect of colloidal gold nanoparticles on the transmittance of glasses with SiO_2_ and TBAB (5% wt.) coating. Colloidal gold nanoparticles with concentrations ranging from 0.9 × 10^−9^ to 4.4 × 10^−9^ mol/mL were introduced into SiO_2_ and TBAB solution. The introduction of gold nanoparticles leads to an increase in transmittance of the samples (annealing temperature 200 °C) (Figure 3a). The dependence of transmittance on the concentration of nanoparticles is extreme. The maximum transmittance of the sample, ~96.5%, is observed at the concentration of colloidal gold nanoparticles of 2.6 × 10^−9^ mol/mL (curve 3), in contrast to ~91.5% for the same coated sample without gold nanoparticles (curve 1). For the concentrations of gold nanoparticles above or below the optimal value, the transmittance of the prepared coatings decreases. At the 250 °C annealing temperature, the transmittance does not depend on the concentration of gold nanoparticles. In the entire range of concentrations studied, the introduction of nanoparticles results in additional increase in transmittance of samples up to ~98.5% (Figure 3b). Transmittance of glass without colloidal gold nanoparticles is ~97% (Figure 3b, curve 1).

According to the literature, gold nanoparticles catalyze the oxidation of organic substrates in the pores of silicon dioxide to CO_2_ and H_2_O at a temperature of ~200 °C. It is assumed that molecular adsorption of oxygen occurs on the surface of gold nanoparticles followed by charge transfer and the formation of surface O^2−^ ions. Adsorbed reactive oxygen promotes fast decomposition of organic additives [25,26,27,28,29,30,31,32].

SEM images of the surface of the coated glasses were obtained using a scanning electron microscope. As examples, the photos of SiO_2_-based coatings with TBAB and TBAB/gold additives obtained at the film annealing temperature of T = 200 °C are presented. Figure 4a,b demonstrates that the coatings are porous and they have a uniform, smooth and crack-free morphology over the entire surface. Histograms of pore area distribution for the samples studied are shown in Figure 4(a1,b1). The histogram for SiO_2_/TBAB coated glass with transmittance of ~92% (Figure 4(a1)) shows that the number of pores up to 4 nm^2^ is about 2500. Furthermore, the coating contains large pores with an area of more than 50 nm^2^ (its quantity is about 100).

The refractive index of the film depends on its air content as follows:
n = (n_air_ n_s_)^½^,(1)
where n, n_air_, n_s_ are the refractive indices of the film, air and substrate, respectively [34,35,36].

Due to the penetration of air with refractive index n = 1 into the nanopores, the refractive index of the coated glass decreases, and the transmittance improves [37,38]. The refractive index of the coated glass should be 1.23 (at 1.51 for the uncoated glass) to obtain resulting material with the transmittance of ~99%. This value corresponds to a film porosity of 0.45–0.50 [18]. Table 1 shows the results of the refractive index measurements and coating thicknesses of the studied samples. At approximately equal thickness, the coated glass containing gold nanoparticles has the higher refractive index than the coated glass without gold. This contradicts the data on the transmittance of these samples. It can be assumed that the higher transmittance of gold-coated glass is due to the increased uniformity of pores in the coatings. As can be seen from the histogram for the SiO_2_/TBAB/gold sample (Figure 4(b1)), the area of most of the pores does not exceed 3 nm^2^, although the SEM image of the coating (Figure 4b) shows pores up to 10 nm^2^ in size.

For comparison, Figure 4c shows the SEM image and histogram of pores of the coating based on SiO_2_ and block copolymer F-127 (3 wt.%) at an annealing temperature T = 400 °C, with ~98% transmittance and n = 1.33 [39]. According to the histogram data (Figure 4(c1)), there are no pores larger than 30 nm^2^, and most of the pores have an area size of up to 4 nm^2^ (their quantity is about 4400), which is comparable with the histogram data for SiO_2_/TBAB/Gold coating (Figure 4(b1)). However, the transmittance of glass with such coating is ~96.5%. The capabilities of the SEM method allow estimating the location of pores on the surface of the coatings only. The transmittance is evaluated through the entire volume of the film, which varies in thickness around 100 nm (Table 1). Since there is no significant difference in the pore sizes on the surface for all coatings, we can assume that the difference in transmittance of 2.5% (from ~96.5 to ~98%) is due to the ordering of pores in the coatings. It can be seen from the SEM images of the surface of the coating based on SiO_2_ and block copolymer F-127 (Figure 4c) that the pores are homogeneous and form a regular macrostructure.

Figure 5 shows the elemental analysis of the SiO_2_/TBAB-based film. All elements are mixed well and distributed uniformly in the film. High signals of the elements Si (27 wt.%) and O (40 wt.%) are observed. The content of C is also observed in the film. It looks like a residual content of the organic additive after annealing. The presence of other elements such as Na, Mg, Al and Ca is a consequence of diffusion of impurities from the glass volume.

## 4. Conclusions

Two complementary approaches to reduce the annealing temperature for the formation of antireflective coatings on the surface of silicate glass are proposed in this work. The first one is the use of the TBAB additive, which significantly reduces the annealing temperature from the traditional 500 °C to 250 °C, allowing to obtain a sample with a transmittance of ~97.5%. The other one is the use of gold nanoparticles in addition to TBAB, which catalyzes the thermal decomposition of TBAB already at 200 °C. The resulting porous structure of the coating decreases the refractive index and increases the transmittance of the sample to ~96.5%. The maximal transmittance of the sample of ~98.5% was achieved at the annealing temperature of 250 °C. The obtained results indicate the prospects of research in this direction for their practical implementation in obtaining antireflective coatings on the surface of silicate glass.

## Figures and Tables

**Figure 1 materials-15-07658-f001:**
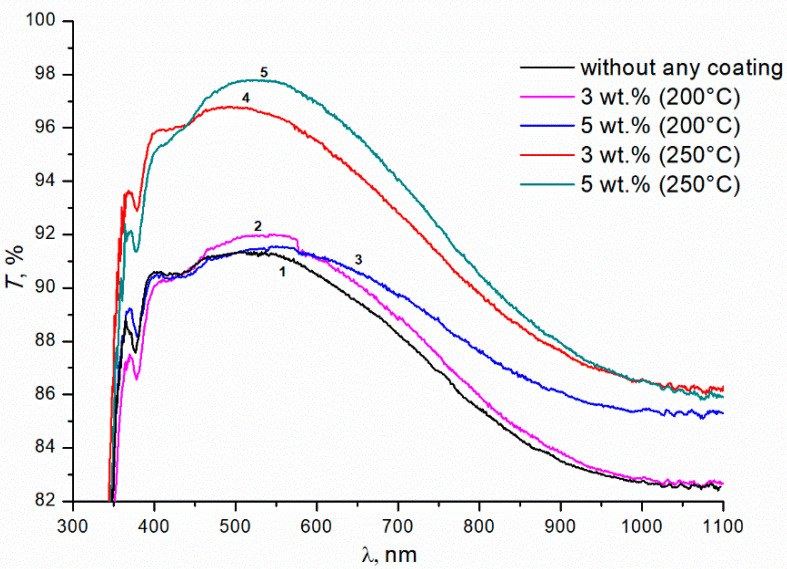
Transmittance of silicate glass without any coating and with the coatings obtained from sols of silicon dioxide with the addition of TBAB of various concentrations (wt.%). Annealing temperatures of gel at 200 °C and 250 °C; duration of annealing is 1 h.

**Figure 2 materials-15-07658-f002:**
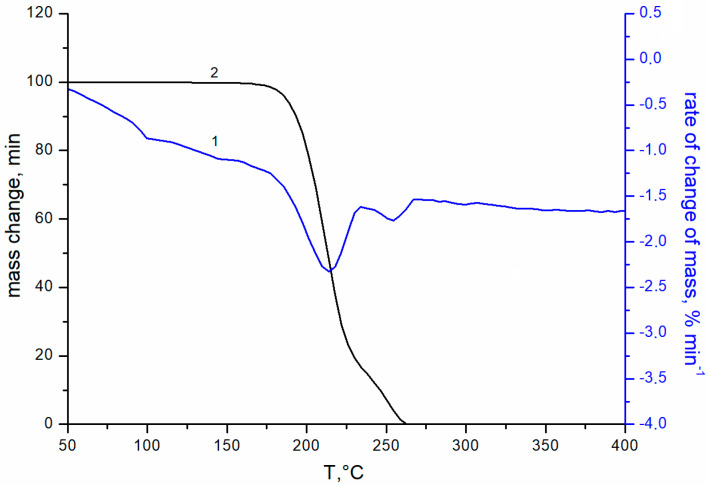
TGA curves of TBAB addition: 1—integral curve; 2—differential curve; heating rate 5 deg min^−1^, flow rate 80 mL min^−1^ in an air atmosphere.

**Figure 3 materials-15-07658-f003:**
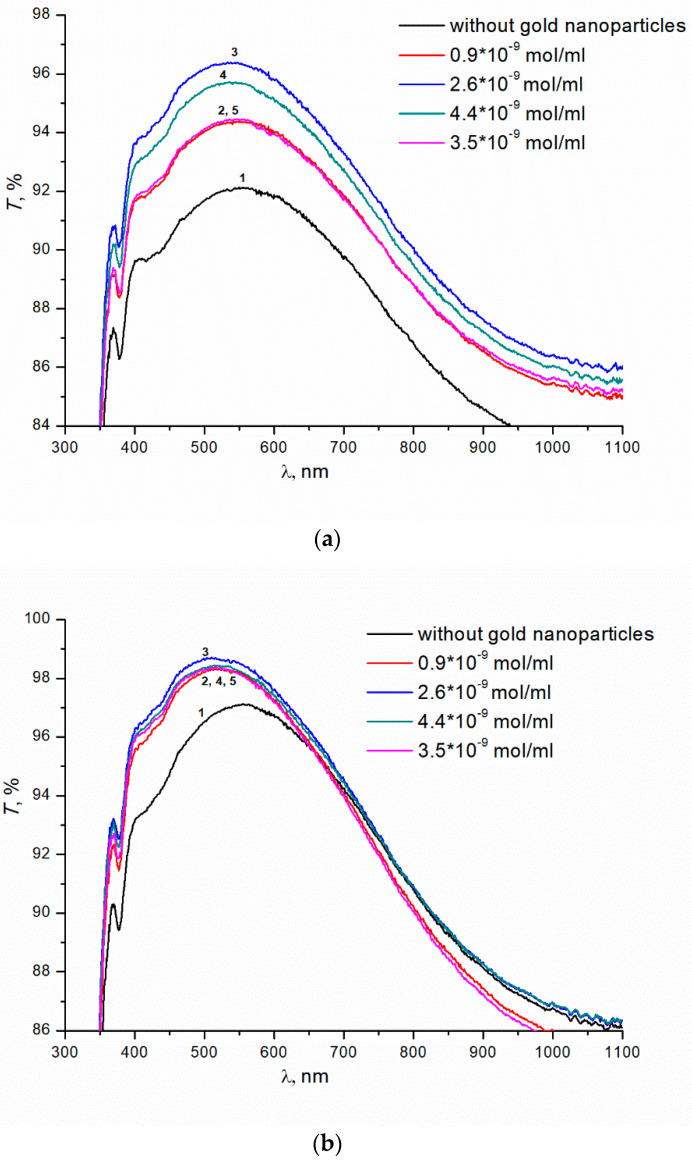
(**a**) Transmittance of silicate glass with the coatings obtained from sols of silicon dioxide with the addition of TBAB (5 wt.%) and gold nanoparticles of various concentrations. Annealing temperatures of gel at 200 °C, 1 h. (**b**) Transmittance of silicate glass with the coatings obtained from sols of silicon dioxide with the addition of TBAB (5 wt.%) and gold nanoparticles of various concentrations. Annealing temperatures of gel at 250 °C, 1 h.

**Figure 4 materials-15-07658-f004:**
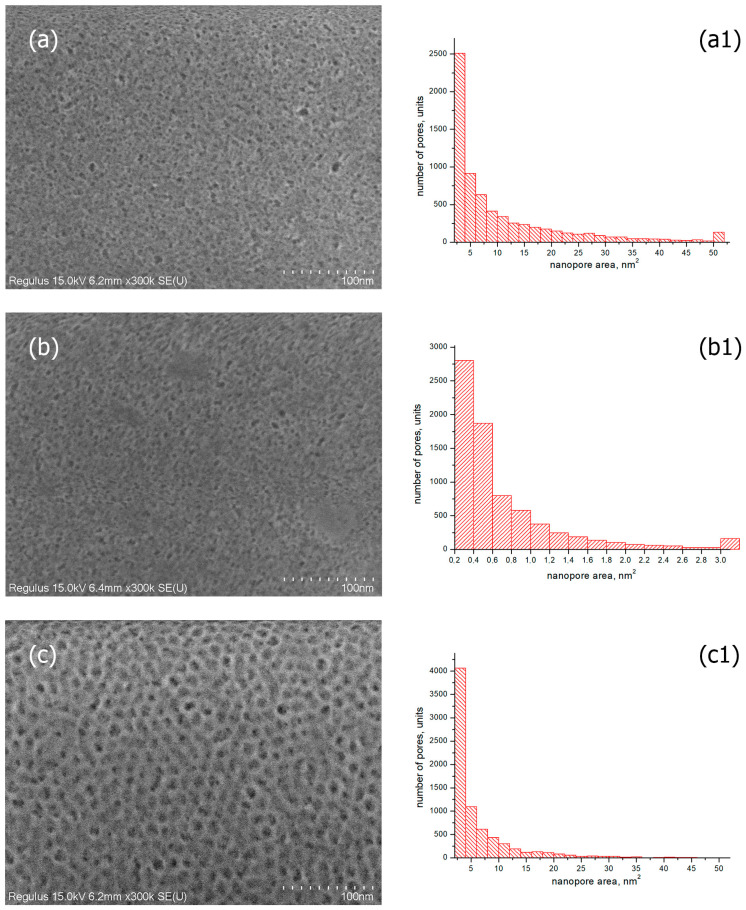
SEM images of silicate glass with the coatings obtained from sols of silicon dioxide with the addition of (**a**) TBAB (5 wt.%), 200 °C; (**b**) TBAB (5 wt.%)/gold nanoparticles 2.6 × 10^−9^ mol/mL, 200 °C; (**c**) F-127 (3 wt.%), 400 °C; (the corresponded images (**a1**–**c1**) are the histograms of particle size distributions).

**Figure 5 materials-15-07658-f005:**
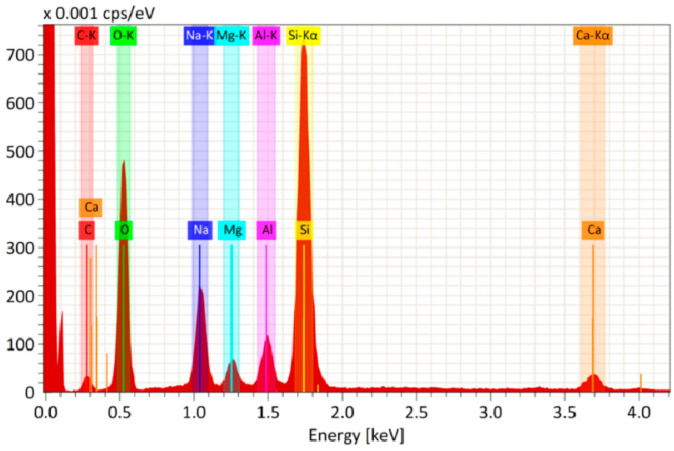
EDS spectrum of elemental analysis of glass with antireflective coating based on SiO_2_/TBAB (5 wt.%)/Gold (2.6 × 10^−9^ mol/mL).

**Table 1 materials-15-07658-t001:** Thickness and refractive index of films based on SiO_2_/TBAB, SiO_2_/TBAB/gold and SiO_2_/F127.

	n	d, nm	*T*, %	T, °C
SiO_2_/TBAB (5 wt.%)	1.38	106.4	~92	200
SiO_2_/TBAB (5 wt.%)/Gold (2.6 × 10^−9^ mol/mL)	1.42	108.5	~96.5	200
SiO_2_/F127 [39]	1.33	112	~98	400

Where **n** are the refractive index of the film, **d** are the thickness of the film, ***T***—is the transmittance of silicate glass with the coatings, **T**—is the annealing temperature of silicate glass with the coatings.

## Data Availability

Not applicable.

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
