# Peer review of "Novel Antireflection Coatings Obtained by Low-Temperature Annealing in the Presence of Tetrabutylammonium Bromide and Gold Nanoparticles"

_materials, 2022, doi:10.3390/ma15217658_

Round 1

Reviewer 1 Report

The following revisions are provided for the revision.

1. There are too much keywords used, while there are problems with semantic repetition (e. g. antireflection coatings, nanocomposite films, and thin films). I would suggest streamlining it and removing any unnecessary repetitive words.

2. The introduction part should be improved. The organization and logic of the introduction is inadequate, which will need some editing to improve the readability. A rewrite is needed.

3. It is recommended that the markings in figure 1, 3, and 4 be marked with a color distinction in the figure, rather than in the figure title, for example, the 2-5 curves in figure 4, which we cannot distinguish. In addition, coating surface morphological in figure 5 should be presented at the same magnification.

4. As seen in Figure 1, the wavelengths of maximum light transmission are different. Does annealing temperature or concentration have an effect on them?​ Please explain properly. Similarly, there is a clear difference between curves 4 and 5, which is believed to be caused by the concentration of TBAB, but the reason is not explained in the manuscript.

5. In Figure 3 and 4, the results of the effect of Au nanoparticles on the light transmission rate can be seen, not only the concentration of Au nanoparticles has an effect on it and annealing temperature also has an effect, in lines 175-179 just explain the role of Au nanoparticles which is not enough, please add to explain this part, at the same time, for this critical argument just cite a reference is lack of convincing.

6. The same problem exists in lines 147-149, where just two sentences are not enough to explain why the coating has a higher light transmission at an annealing temperature of 250°C. Perhaps further explanation can be given in terms of chemical reactions or Gibbs free energy to illustrate the content.

7. It is believed that the subject of the manuscript is that TBAB and Au nanoparticles at 200-250°C annealing temperature affect the pore structure of the coating and thus the light transmission. However, there is less discussion on the influence mechanism in the whole manuscript, and a more clear explanation is needed.

8. At last, the authors will also need to have the manuscript undergo a substantial grammar check.

Author Response

  1. There are too much keywords used, while there are problems with semantic repetition (e. g. antireflection coatings, nanocomposite films, and thin films). I would suggest streamlining it and removing any unnecessary repetitive words.

Answer: corrected

  1. The introduction part should be improved. The organization and logic of the introduction is inadequate, which will need some editing to improve the readability. A rewrite is needed.

Answer: corrected

  1. It is recommended that the markings in figure 1, 3, and 4 be marked with a color distinction in the figure, rather than in the figure title, for example, the 2-5 curves in figure 4, which we cannot distinguish. In addition, coating surface morphological in figure 5 should be presented at the same magnification.

Answer:corrected

  1. As seen in Figure 1, the wavelengths of maximum light transmission are different. Does annealing temperature or concentration have an effect on them? Please explain properly. Similarly, there is a clear difference between curves 4 and 5, which is believed to be caused by the concentration of TBAB, but the reason is not explained in the manuscript.

Answer:It is known that the wavelength at which the maximumoptical transmission is observed is directly proportionalto the refractive index and thickness of a single-layerantireflection coating (λ = 4nd). With increasingconcentration of the concentration of TBABin solution, theviscosity of the latter grows and the content ofTBAB in a film becomes higher, which leads, in the end,to an increase in the thickness of a coating produced by the dip-coating method. Thus, the maximum opticaltransmission shifts to longer wavelengths with increasing TBAB concentration in solution.

The optimal additive content in the solution is determined by the concentrationsratio of silicon dioxide and additive. As the additive content in the solution increases, we assume that the size of nanopores and their total volume should increase in the coating after annealing. The refractive index decreases accordingly and the optical transmittance of coated glass increases (it was added to the article).

  1. In Figure 3 and 4, the results of the effect of Au nanoparticles on the light transmission rate can be seen, not only the concentration of Au nanoparticles has an effect on it and annealing temperature also has an effect, in lines 175-179 just explain the role of Au nanoparticles which is not enough, please add to explain this part, at the same time, for this critical argument just cite a reference is lack of convincing.

Answer:We discuss the effect of annealing temperature on the transmittance of coated glass in Figure 1 at the beginning of this article.

We introduce gold nanoparticles to see if we can improve the result (Figure 1) at these annealing temperatures. Therefore, in Figures 3a and 3b we consider the effect of only the concentration of gold nanoparticles. Since at 250°C the additive is completely removed even without the introduction of gold nanoparticles, the light transmission of the coated glass is therefore almost unchanged when the nanoparticles are added (Figure 3b).

  1. The same problem exists in lines 147-149, where just two sentences are not enough to explain why the coating has a higher light transmission at an annealing temperature of 250°C. Perhaps further explanation can be given in terms of chemical reactions or Gibbs free energy to illustrate the content.

Answer:Probably, at a temperature of 200°C, TBAB remains in the coating, which is confirmed by the thermogravimetric analysis (Figure 2). The temperature of 200°C corresponds to the initial stage of TBAB decomposition. However, at an annealing temperature of 250°C the additive completely decomposes (Figure 2), which leads to a significant increase in the light transmission of the sample (it was added to the article).

TBAB decomposes at temperature above 200 °C. Most likely, the Hoffmann dealkylation occurs in this case. It is known that these are the base-catalyzed reactions. In the present work, we don’t analyze the products of the thermal decomposition in extremely small amounts, but we assume that TBAB decays to form mainly tributylamine (TBA). This compound begins to boil at temperature ≈ 200 °C, and it completely evaporates at 250°C. [WeisshaarD.E., Earl G.W.,AmolinsM.W.,Mickalowski K.L.,Norberg J.G.,Rekken B.D., Burgess A.M., KaemingkB.D., Behrens K.C., Investigation of the Stability of Quaternary Ammonium MethylCarbonates, J SurfactDeterg2012 (15), pp.199–205; DOI 10.1007/s11743-011-1292-1]

  1. It is believed that the subject of the manuscript is that TBAB and Au nanoparticles at 200-250°C annealing temperature affect the pore structure of the coating and thus the light transmission. However, there is less discussion on the influence mechanism in the whole manuscript, and a more clearexplanation is needed.

Answer:The mechanism of TBAB decomposition can proceed in at least two ways:

  • The Hoffmann dealkylation. Without gold nanoparticles.
  • Gold catalyzes redox reactions, therefore lowering the activation energy of the process. We have not studied the kinetics of luminescence, but lowering the process temperature by 50°C (from 250°C to 200°C) may correspond to an activation energy of 10-15 kJ/mol.
  1. At last, the authors will also need to have the manuscript undergo a substantial grammar check.

Answer: corrected

Reviewer 2 Report

Review report

Manuscript title: Novel Antireflection Coatings Obtained by Low-Temperature  Annealing in the Presence of Tetrabutylammonium Bromide  and Gold Nanoparticles

Manuscript ID: materials-1971533

Review comments: The work is interesting and it has a very good novelty. The work demonstrates about the Novel Antireflection Coatings Obtained by Low-Temperature  Annealing in the Presence of Tetrabutylammonium Bromide and Gold Nanoparticles. I think it will be published in the current journal after modification the issues raised. 

1.                  In the abstract there are mixing of present and past form of tenses. Its totally wrong. It should be one particular form.

2.                   “Keywords: silicate glass, sol compositions, silicon dioxide, antireflection coatings, nanocomposite 21 films, thin films, tetrabutylammonium bromide, gold nanoparticles, optical properties”-there is a wrong choosing of words. One word should be used as one word.

3.                  In Introduction part there needs more references for comparing.

4.                  In the discussion part need more references for comparison to others works.

5.                  There are many grammatical mistakes. The English needs to be rechecked and make it more corrections.

Author Response

RESPONSE TO THE REVIEWER COMMENTS

We would like thank the Reviewer for the valuable comments, which helped us make our manuscript better.

Reviewer 2:The work is interesting and it has a very good novelty. The work demonstrates about the Novel Antireflection Coatings Obtained by Low-Temperature Annealing in the Presence of Tetrabutylammonium Bromide and Gold Nanoparticles. I think it will be published in the current journal after modification the issues raised.

1.In the abstract there are mixing of present and past form of tenses. It’s totally wrong. It should be one particular form.

Answer: Abstract has been re-written according to the comment of the Reviewer.

2.“Keywords: silicate glass, sol compositions, silicon dioxide, antireflection coatings, nanocomposite 21 films, thin films, tetrabutylammonium bromide, gold nanoparticles, optical properties”-there is a wrong choosing of words. One word should be used as one word.

Answer: corrected

3.In Introduction part there needs more references for comparing

Answer:corrected

  1. In the discussion part need more references for comparison to others works.

Answer:corrected

5.There are many grammatical mistakes. The English needs to be rechecked and make it more corrections.

Answer:corrected

Reviewer 3 Report

The authors report on the transmittance of anti-reflection coatings produced by sol-gel method on silicate glass. Addition of tetrabutylammonium bromide (TBAB) and Au nanoparticles to the coatings on their properties is investigated and compared with a reference coating produced without the above substances. In addition, the effect of annealing temperature was investigated at 200 and 250°C.

The manuscript contains interesting results, which would be worth to be published.

Nevertheless, the authors should address the following points prior to publication.

1) The authors do not communicate the transmittances of all coatings at one place, so the comparison of the values is sometimes difficult.

For example,

In the abstract, the transmittances of the reference coating of ~ 91.5 % is not mentioned.

In the abstract, the transmittance of the coating with 5 wt.% TBAB is stated to be ~ 98.0 %, while in the main text, page 4, a transmittance is stated to be ~ 97.5 %.

In the abstract, the transmittance of the coating achieved by 5 wt.% TBAB, 2.6 x 10-9 mol/ml Au nanoparticles, and annealing is stated to be 96.5 %. However, no reference value is stated, so the achievement cannot be appreciated.

Please communicate the reference values and the experimental values of the transmittances together for better comparison.

Please stay consistent with stating the transmittance values throughout the manuscript.

2) Introduction

The introduction section is written as one paragraph.

Please consider dividing the section into several paragraph based on the focus, for example, introduction of the sol-gel method, previous results with CTAB, introduction of the goal of the manuscript.

3) Introduction

“Using if TBAB can lead to a change in the pore structure of the antireflection coating so that the antireflection effect is increased at a low annealing temperature of 200-250°C.”

This statement essentially concludes the results of the manuscript.

Please consider to skip the conclusion in the introduction.

4) Page 3

Acceleration voltages of 2 and 15 kV were used for imaging and compositional analysis of the coatings in the SEM / EDS.

Were the measurements possible without charging?

Please explain.

5) Fig. 2 shows the results of the thermogravimetry measurements – the absolute mass change and the rate of the mass change in % per minute.

Please add the information about the heating rate in the figure caption as the heating rate is crucial for stating the rate of the mass in change in % per minute.

6) Figs. 3 and 4 show the transmittances of the coating at 200 and 250°C.

Since, the data are of similar nature, please consider presenting the data as Fig. 3(a) and 3(b).

In addition, is it difficult to distinguish the curve 2 and 5 in Fig. 3. Similarly, it is difficult to distinguish the curves in Fig. 4.

Moreover, in the figure caption of Fig. 3, curve 1 is referred to a coating without gold nanoparticles, while in the figure caption of Fig. 4, curve is 1 referred to a coating without Au nanoparticles.

Please improve readability of the figures if possible.

Please unify the style in writing - gold or Au.

7) Fig. 5

The magnification and the scale bar are barely visible in the figure.

Moreover, the comparison of the samples is limited due to different magnification of the SEM photos.

Please consider to present SEM photos with the same magnification, if possible.

8) Conclusions

In the conclusion, “halving” of the temperature from 500 to 250°C for synthesis of samples with a high transmittance upon utilization of TBAB is reported.

The term “halving” is mathematically correct, however, it would not hold, when the temperature would be expressed in other temperature units, such as the thermodynamic temperature in K.

Please consider re-phrasing.

Author Response

We would like thank the Reviewer for the valuable comments, which helped us make our manuscript better.

Reviewer 3:The authors report on the transmittance of anti-reflection coatings produced by sol-gel method on silicate glass. Addition of tetrabutylammonium bromide (TBAB) and Au nanoparticles to the coatings on their properties is investigated and compared with a reference coating produced without the above substances. In addition, the effect of annealing temperature was investigated at 200 and 250°C.

The manuscript contains interesting results, which would be worth to be published.

Nevertheless, the authors should address the following points prior to publication.

1) The authors do not communicate the transmittances of all coatings at one place, so the comparison of the values is sometimes difficult.

For example,

In the abstract, the transmittances of the reference coating of ~ 91.5 % is not mentioned.

In the abstract, the transmittance of the coating with 5 wt.% TBAB is stated to be ~ 98.0 %, while in the main text, page 4, a transmittance is stated to be ~ 97.5 %.

In the abstract, the transmittance of the coating achieved by 5 wt.% TBAB, 2.6 x 10-9 mol/ml Au nanoparticles, and annealing is stated to be 96.5 %. However, no reference value is stated, so the achievement cannot be appreciated.

Please communicate the reference values and the experimental values of the transmittances together for better comparison.

Please stay consistent with stating the transmittance values throughout the manuscript.

Answer: corrected

  1. Introduction

The introduction section is written as one paragraph.

Please consider dividing the section into several paragraph based on the focus, for example, introduction of the sol-gel method, previous results with CTAB, introduction of the goal of the manuscript.

Answer: corrected

  1. “Using if TBAB can lead to a change in the pore structure of the antireflection coating so that the antireflection effect is increased at a low annealing temperature of 200-250°C.”

This statement essentially concludes the results of the manuscript.

Please consider to skip the conclusion in the introduction.

Answer:corrected

  1. Page 3

Acceleration voltages of 2 and 15 kV were used for imaging and compositional analysis of the coatings in the SEM / EDS.

Were the measurements possible without charging?

Please explain

Answer:At the beginning of the research acceleration voltage of 2 kV was used to antireflection coatings SEM analysis. Unfortunately this mod has lead to low resolution images. Later it was found that the increase of acceleration voltage up to 15 kV allows to obtain high quality micrographs. In addition, such an accelerating voltage was necessary for a reliable determination of gold atoms by the EDS method. We have removed the reference to the 2 kV accelerating voltage in the manuscript so that the reader does not get confused.

  1. Fig. 2 shows the results of the thermogravimetry measurements – the absolute mass change and the rate of the mass change in % per minute.

Please add the information about the heating rate in the figure caption as the heating rate is crucial for stating the rate of the mass in change in % per minute.

Answer:corrected

  1. Figs. 3 and 4 show the transmittances of the coating at 200 and 250°C.

Since, the data are of similar nature, please consider presenting the data as Fig. 3(a) and 3(b).

In addition, is it difficult to distinguish the curve 2 and 5 in Fig. 3. Similarly, it is difficult to distinguish the curves in Fig. 4.

Moreover, in the figure caption of Fig. 3, curve 1 is referred to a coating without gold nanoparticles, while in the figure caption of Fig. 4, curve is 1 referred to a coating without Au nanoparticles.

Please improve readability of the figures if possible.

Please unify the style in writing - gold or Au.

Answer: corrected

  1. Fig. 5

The magnification and the scale bar are barely visible in the figure.

Moreover, the comparison of the samples is limited due to different magnification of the SEM photos.

Please consider to present SEM photos with the same magnification, if possible.

Answer: corrected

  1. Conclusions

In the conclusion, “halving” of the temperature from 500 to 250°C for synthesis of samples with a high transmittance upon utilization of TBAB is reported.

The term “halving” is mathematically correct, however, it would not hold, when the temperature would be expressed in other temperature units, such as the thermodynamic temperature in K.

Please consider re-phrasing.

Answer: corrected

Round 2

Reviewer 1 Report

The authors have made thorough modifications to their manuscript, and correctly addressed all the referee's remarks. I believe the current version deserves rapid publication in Materials. I believe that this is a valuable work for the antireflective coatings industry. However, there are a few places with remaining minor issues, such as punctuation, and a few slips of the pen and so on. It is strongly recommended to check the whole manuscript more carefully again.